

# Lead transfer in the soil-root-plant system in a highly contaminated Andean area

Jorge Castro-Bedriñana[1], Doris Chirinos-Peinado[2], Edgar Garcia-Olarte[3] and Rolando Quispe-Ramos[3]

[1] Specialized Research Institute of the Faculty of Zootechnics, Universidad Nacional del Centro del Perú, Huancayo, Junín, Perú
[2] Nutritional Food Safety Research Center, Universidad Nacional del Centro del Perú, Huancayo, Junín, Perú
[3] Faculty of Zootechnics, Universidad Nacional del Centro del Perú, Huancayo, Junín, Perú

## ABSTRACT

Lead (Pb) is highly toxic heavy metal that is detrimental to the food system. There are large mining and metallurgical companies in the central highlands of Peru that have been active for almost a century and contribute to air, water, and soil pollution, affecting food quality and causing damage to the environment and human health. Our study, conducted in 2018, assessed the content and transfer of lead in the soil-root-plant system in the high Andean grasslands in a geographical area near the metallurgical complex of La Oroya. Lead levels were measured in 120 samples of top soil (0–20 cm), roots, and grass shoots by flame atomic absorption spectroscopy. No significant differences were found between the soil pH, organic matter content, and lead among the samples evaluated ($P > 0.05$). Mean Pb concentrations decreased in the order of soil > root > shoot ($P < 0.01$) ($212.36 \pm 38.40$, $154.65 \pm 52.85$ and $19.71 \pm 2.81$ mg/kg, respectively). The soil-to-root Pb bioconcentration factor, root-to-shoot translocation factor, and soil-to-shoot bioaccumulation factor values were $0.74 \pm 0.26$, $0.14 \pm 0.06$ and $0.10 \pm 0.03$, respectively. Lead in the soil was 3.03 times higher than the maximum limit for agricultural soil, and was 1.97 times higher than the value limit for fodder. Our findings are important and show that soils and pasture in this geographical area have high Pb levels due to metallurgical emissions that have been occurring since 1922. Such pollution negatively impacts health and the socio-economic status of the exposed populations.

Corresponding author
Jorge Castro-Bedriñana,
jcastro@uncp.edu.pe,
jorgecastrobe@yahoo.com

## INTRODUCTION

Lead (Pb) is highly toxic to living organisms; it is naturally present in the earth's crust in harmless concentrations, usually between 15–40 mg/kg (*Fahr et al., 2013*; *Amari, Ghnaya & Abdelly, 2017*).

According to the European REACH regulation and the Agency for Toxic Substances and Disease Registry, Pb is the second most harmful contaminant to humans after arsenic (*ATSDR, 2007*; *Pourrut et al., 2011*).

Pb is widely used in industry (*Wuana & Okieimen, 2011*) and its metallurgical process emits fine particulate material that travels many kilometers through the air (*Martin et al., 2017*) depositing in water, soil and when it exceeds certain limits it geoaccumulates, bioaccumulates and biomagnifies (*Lokeshwari & Chandrappa, 2006*), and can reach toxic levels with serious consequences for ecosystems (*Alloway, 2013*; *Kong, 2014*).

Anthropogenic Pb released into the atmosphere is deposited and accumulated in the upper soil layer and, due to its long biological half-life and high bioaccumulation potential, is absorbed by plant roots (*Nascimento et al., 2014*; *Hou et al., 2014*), generating Pb-laden foods that are harmful to human health (*Li et al., 2005*; *Li et al., 2017*; *Castro Bedriñana, Chirinos Peinado & Ríos Ríos, 2016*; *Martin et al., 2017*), especially for infants and children (*Chirinos-Peinado & Castro-Bedriñana, 2020*).

Soil quality plays an important role in food safety by determining the possible composition of forage at the early levels of the food chain (*Tóth et al., 2016*); so, analysis of the top of the soil (typically the top 0–20 cm) is valuable in assessing heavy metal contamination in grasslands (*Martin et al., 2017*).

Soil contamination and its potential impact on human health has not been extensively studied in Peru, especially in the high Andean areas where mining and metallurgical activities are carried out in addition to livestock activities (*Chang Kee et al., 2018*).

The La Oroya polymetallic mining-metallurgical complex, in operation since 1922, generates emissions and acid fumes that exceed international standards and contaminate the surrounding ecosystems (*Alvarez-Berríos et al., 2016*), so much so that La Oroya is now the fifth most contaminated place on the planet (*Blacksmith Institute, 2007*) and in these almost 100 years the soils and vegetation of central Peru have been contaminated by heavy metals and toxic substances (*Alvarez-Berríos et al., 2016*; *USDA, 2017*).

A recent study shows that milk produced in an area close to the metallurgical industry has quantities of Pb and Cd above the maximum permitted levels (*Chirinos-Peinado & Castro-Bedriñana, 2020*), which makes soil quality an issue of economic and social importance (*Kong, 2014*) not only for Peru but for the world.

The threshold of Pb content in agricultural soils and forages have wide variations (*Chen et al., 2018*). Bulgaria, Denmark, Russia, Thailand, Norway, Finland and the Czech Republic set the lowest values (40, 40, 55, 55, 60, 60 and 60 mg/kg), while Switzerland, Belgium, the Republic of Korea and the Netherlands set the highest values (1,000, 700, 600 and 530 mg/kg).

In Peru, the Ministry of the Environment establishes an Environmental Quality Standard for agricultural soils of 70 mg/kg (*MINAM, 2017*).

Many studies on heavy metal contamination have been generated in controlled studies with pre-set amounts of samples (*Adamse, Van Der Fels-Klerx & De Jong, 2017*); therefore, a strength of this ecological study is its in-situ design over time.

Information on Pb contamination in the soil-root-plant system in an area affected by metallurgical emissions in the Andean region generates scientific evidence for monitoring Pb contamination in the soil and its transfer to the food chain base.

There are reports that the absorption and bioaccumulation of heavy metals in plants depends, among other factors, on the plant species (*Sharma & Dubey, 2005*) and seasonal variations (*Bidar et al., 2009*; *Yabanli, Yozukmaz & Sel, 2014*), this study has determined the concentrations and transference of Pb in the soil-root-blossom system of natural and cultivated pastures in samples collected during the rainy and dry seasons, well defined seasons in the central Andes, in an area of cattle, sheep and camelid farming in South America, located 20 km from the largest polymetallic metallurgical complex in this region of the world.

The Pb content of the soil was compared with the Peruvian standard of the Ministry of Environment and due to the existence of a national standard, the Pb of the forage was compared with an international standard.

## MATERIALS AND METHODS

### Study approval

The study was approved by the Evaluation Committee of the Research General Institute of the Universidad Nacional del Centro del Perú, study which is part of a larger project (Project number: 1565-R-2917-UNCP). Sampling of soil and grasslands was approved by the President of the Peasant Community "Purísima Concepción de Paccha", Yauli, La Oroya.

### Description of the study site

The research was carried out in the pastures of the Paccha Peasant Community, located in the central Andes of Peru, department of Junin, province of Yauli (8 724 632.8 N, 417 610.7 E; 3,750 m a.s.l., minimum and maximum temperature of −3.1 and 18.2 °C) that covers 16,564 ha of grassland, of which 13.73 ha are irrigated cultivated pastures.

The study site is located 20 km from the Polymetallic Metallurgical Complex of La Oroya, and is subject to particulate emissions that contaminate the ecosystem from the beginning of its activities. Figure 1 shows the location map and the distribution of the sampling points for cultivated and natural soils and pastures in April (post-rainfall) and September (post-dry) 2018.

This area was chosen because it is representative of the communities near La Oroya, and because of their agro-ecological conditions they are dedicated to raising cattle, sheep, and alpacas, having approximately 11 ha (hectares) of natural pastures composed mainly of *Festuca dolichophylla*, *Bromus catharticus*, *Bromus lanatus*, *Nasella meyeniana*, *Calamagrostis heterophylla*, *Piptochaetium faetertonei*, *Nasella publiflora*, *Asiachnae pulvinata*, *Margaricarpus pinnatus*, *Oenothera multicaulis*, *Trifolium amabile* and 6 ha of association *Lolium perenne* and *Trifolium repens* installed 15 years ago, characterized by a poor condition.

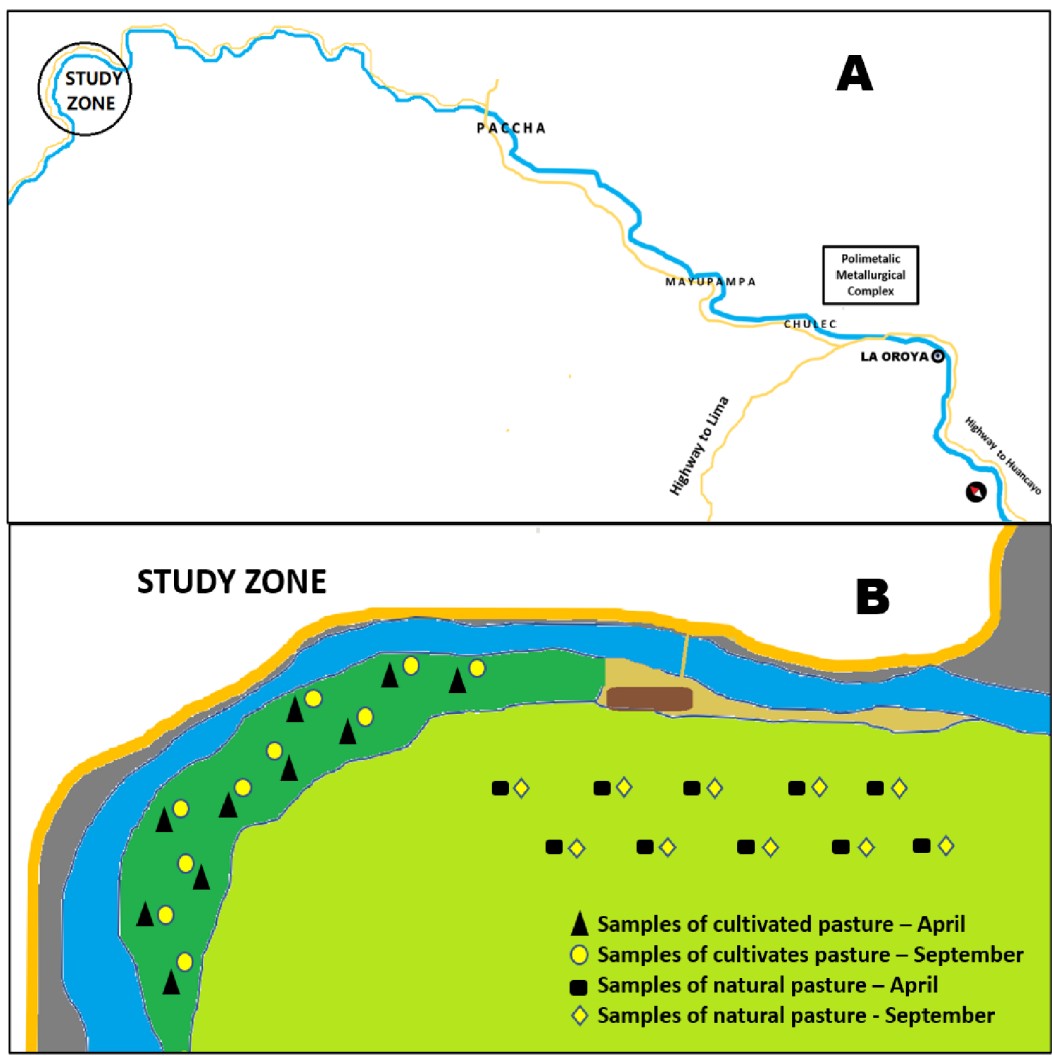

**Figure 1 Partial map of La Oroya-Peru, research area (3,900–4,500 m. a.s.l).** La Oroya-Peru research area partial map (3900-4500 m asl). (A)The road from the La Oroya mining-metallurgical complex to the study area (20 km). (B) The study site and the distribution of soil and pasture sampling points (1 m2/sample) in two periods.

The quantification of Pb in the soil, roots and shoots was made by means of flame atomic absorption spectroscopy (FLAA) in the laboratory Baltic Control SAC, accredited by the National Institute of Quality of Peru.

Contamination of the study area from emissions from the La Oroya mine/metallurgical complex is supported by the 2005 air dispersion modeling study of La Oroya, which highlights the relative importance that emissions from the main stack increase as the distance from the complex increases, while fugitive sources smaller than the main stack are responsible for most local impacts. The average wind speed is 5.76 km/h, with a predominantly northeast direction for different times of the year. The wind transports the

fine particle material to surrounding communities, including Paccha (*Ramírez & Corcuera, 2015*) where it was conducted in the study.

## Sampling procedures

Topsoil sampling (0–20 cm depth) was carried out at 20 points in the natural grassland area and 20 points in the cultivated grassland area (Fig. 1), which were collected in April and September 2018. A total of 120 samples (soils: 40, roots: 40, shoots: 40) were collected using standardized sampling procedures (*Tóth et al., 2016*; *Martin et al., 2017*).

From the same sampling site soil and grasses of 1 $m^2$ were sampled with a stainless-steel shovel, taking approximately 0.5 kg of soil from each sample point and the grasses present (*Tóth, Jones & Montanarella, 2013*). The grasses were divided into root and aerial parts (*Castro-Bedriñana, Chirinos-Peinado & Peñaloza Fernández, 2020*), and all samples were placed in first-use polyethylene bags with zippered closure to be taken to the laboratory.

## Laboratory analysis

After natural drying for one day, the soil samples were crushed and screened with a two mm thick mesh to remove gravel, stone, and other materials (*Martin et al., 2017*). A portion of this sample was used to estimate the percentages of sand, silt and clay, the rest was pulverized and sieved in 100 mesh, obtaining a fine and homogeneous powder for analysis.

The organic matter in the soil was analyzed by the Walkley-Black method (*Gessesse & Khamzina, 2018*) and the pH by the EPA 9045D of soil and residues (1:2.5 soil to water, w:w) using a model Orion 720A pH/mV/temperature meter.

Roots and shoots were washed with abundant tap water to remove soil particles and rinsed three times with deionized water (*Bidar et al., 2009*), then dried at 70 °C and finely ground (*Ramsumair, Mlambo & Lallo, 2014*).

The samples were digested using the USEPA 3050B method (SW-846). Repeated additions were made of a mixture of concentrated nitric acid ($HNO_3$) and hydrogen peroxide ($H_2O_2$) to one gram of dry sample. Hydrochloric acid (HCl) was added to the initial digestate and the sample was refluxed. The digestate was diluted to a final volume of 100 ml (*USEPA, 1996*).

To quantify the Pb concentration of the different samples, the standard analysis procedure was followed (*Tóth, Jones & Montanarella, 2013*), using a flame atomic absorption spectrophotometer AAS (NAMBEI AA320N), following the protocol of the Official Method 975.03 of the AOAC (*AOAC, 1990*; *USEPA, 1996*).

To ensure analytical precision, the blank method, duplicate samples, and the high- and low-range control standard were used for every 15 samples analyzed. The Sigma-Aldrich Pb 986 ± 4 mg/kg standard was used for the calibration curve. The detection limit of Pb was 0.2 mg/kg.

The units of concentration of Pb were expressed in mg/kg. The Environmental Quality Standard of 70 mg/kg (*MINAM, 2017*) was used to evaluate the Pb concentration in soil and the maximum limit of 10 mg/kg dry matter for forage (*Boularbah et al., 2006a*; *Boularbah et al., 2006b*; *Kabata-Pendias & Mukherjee, 2007*).

**Table 1  Average and range of pH, organic matter (OM) and Pb content in the soil in an area near the polymetallic smelter in the central Andean region of Peru.**

| Period | pH | OM (%) | Pb (mg/kg) |
|---|---|---|---|
| Dry | 6.34 [5.03–7.47] | 3.89 [3.15–4.57] | 206.91 [125.18–269.93] |
| Rain | 6.38 [5.08–7.73] | 3.93 [3.11–4.53] | 217.81 [131.76–284.13] |
| Average | 6.36 [5.03–7.73] | 3.91 [3.11–4.57] | 212.36 [125.18–284.13] |

**Notes.**

There are no statistical differences between pH, MO and Pb average values per period ($P > 0.05$).

## Lead transfer in the soil-root-plant system

The bioaccumulation factor (BF) represents the potential capability of heavy metals' transmission from soil to the edible parts of vegetable (*Hu et al., 2017*). The equation used to estimate the BF of Pb from soil to shoot were the following:

$$BF = [Pb\ Shoot]/[Pb\ soil].$$

A plant's ability to accumulate metals from soils can be estimated using the bioconcentration factor (BCF) (*Yoon et al., 2006*), which is defined as the ratio of metal concentration in the roots to that in soil (*Zou et al., 2011*). A plant's ability to translocate metals from the roots to the shoots is estimated using the translocation factor (TF) which is the ratio of metal concentration in the shoots to that in the roots (*Yanqun et al., 2005*; *Yoon et al., 2006*; *Zou et al., 2011*). The equations used to estimate the BCF and TF were the following:

$$BCF = [Pb\ root]/[Pb\ soil]$$
$$TF = [Pb\ Shoot]/[Pb\ Root]$$

## Statistical analysis

To determine statistical differences between Pb concentrations in soil, roots and shoots by species and samples period, two-way analysis of variance (ANOVA) and Tukey comparisons were performed with a confidence level of 0.95 ($P < 0.05$), using SPSS version 23.0. To compare the Pb level in soil and shoots with the maximum allowable limits (soil: 70 mg/kg; shoot: 10 mg/kg), a single sample "t" tests was performed.

# RESULTS

## Soil composition and Pb content

In general, the soil in the study area is classified as aqueous inceptisol. On average it contains 53% sand, 29% silt and 18% clay, which characterizes a sandy loam soil (*USDA, 2017*), with an organic matter content of 3.9% (Table 1).

No significant differences were found between soil pH per sampling period and per type of pasture ($P > 0.05$). The pH range was between 5.03 and 7.73, indicating that the soils evaluated are weakly acidic to neutral.

**Table 2** Lead concentration in soil, roots and shoots of high Andean pastures (mg/kg) in an area near the polymetallic smelter in the central Andean region of Peru ($n = 40$).

|  | Average | S. D | 95% Confidence Interval | | Minimum | Maximum |
|---|---|---|---|---|---|---|
|  |  |  | Lower limit | Upper limit |  |  |
| Soil | 212.36 a | 38.40 | 200.08 | 224.64 | 125.18 | 284.13 |
| Root | 154.65 b | 52.85 | 137.75 | 171.56 | 77.04 | 263.61 |
| Shoot | 19.71 c | 2.80 | 18.81 | 20.60 | 13.83 | 23.88 |

Notes.
   a, b, c Average Pb contents: soil > root > shoot, vary statistically ($P < 0.01$).

**Table 3** Lead transfer factors in the soil-root-shoot system in high Andean pastures in an area near the polymetallic smelter in the central Andean region of Peru.

| Lead transfer | Average | S. D | 95% Confidence interval | | Minimum | Maximum |
|---|---|---|---|---|---|---|
|  |  |  | Lower limit | Upper limit |  |  |
| BCF Soil-Root | 0.743a | 0.261 | 0.659 | 0.827 | 0.31 | 1.43 |
| TF Root-Shoot | 0.144b | 0.057 | 0.126 | 0.162 | 0.06 | 0.29 |
| BF Soil-Shoot | 0.096b | 0.025 | 0.088 | 0.104 | 0.06 | 0.18 |

Notes.
   a, b: BCF, TF and BF averages whit different letters vary statistically ($P < 0.01$).

## Lead concentration in soil, roots and shoots

The range and average concentration of Pb in the soil, roots and shoots of high Andean grasses are presented in Table 2. These concentrations did not vary between pasture types and sampling periods, so the results are shown globally ($n = 40$). The order of mean Pb concentrations was soil >root >shoot ($P < 0.01$).

The mean Pb concentrations in soil and edible part of pastures are considerably higher ($P < 0.01$) than the maximum limits for agricultural soils (*MINAM, 2017*) and cattle fodder (*Boularbah et al., 2006b*; *Kabata-Pendias & Mukherjee, 2007*)

## Lead transfer in the soil-root-shoot system

The BCF of Pb from soil to root was 5.16 times greater than the TF of Pb from root to shoot and 7.14 times greater than the BF of Pb from soil to shoot ($P < 0.01$), a result that shows that the roots of high Andean pastures produced in a highly contaminated area of the central Andean region bioaccumulate and retain the greatest amount of Pb from the plant (Table 3).

In the current study, of 100% of the Pb of the plant, on average 16% is found in the shoots and 84% is concentrated in the root, restricting the translocation of Pb to the edible part of the grass.

# DISCUSSION

## Physical-chemical characteristics of the soil

The content of silt, clay and sand in the study area was similar to that reported by *Ramírez & Corcuera (2015)* in the soils of Paccha (52% sand, 33% silt and 15% clay), being classified as aqueous inceptisol, with a sandy loam texture. Similar results are reported in Mantaro

Valley, Apata-Jauja, 80 km from La Oroya where soils containing 47, 24 and 29% sand, silt and clay, whose texture is sandy loam - clay loam - sandy loam, with neutral to slightly acidic pH (*Orellana et al., 2019*).

In this study, OM and Pb contents in the soil were similar by type of grassland and sampling period ($P > 0.05$), so the degree of solubility and bioavailability of Pb for plants would also be similar throughout the year (*Lokeshwari & Chandrappa, 2006*) in chronic contamination case.

In the study area the pH was slightly acidic to neutral and the availability of Pb would be similar at the different sampling sites where the OM levels were also similar (Table 1).

The results of the study are indicative of the low OM content of Andean soils in central Peru, and due to the high and persistent toxicity of Pb improvements should be made so that the soil has at least 5% OM (*Enya et al., 2020*) recommending fertilization with high Andean cattle manure.

## Lead concentration in soil, roots and shoots

The soils of La Oroya have been contaminated since the smelter began operations in 1922 (*Chirinos-Peinado & Castro-Bedriñana, 2020*). For 2007 and 2008, in La Oroya and Paccha (study area), an average concentration of PM $_{2.5}$ particles of 32.4 and 20.3 ug/m$^3$ was reported, values that exceed the Environmental Quality Standard (EQS = 15 ug/m$^3$ ) and for PM$_{10}$ of 52.3 and 42. 4 ug / m$^3$ values similar to the EQS = 50 ug/m$^3$) (*MINAM, 2008*), which would be deposited in the soil and most likely assimilated by plants and animals, which could affect human health at the end of the food chain (*Kryshna & Mohan, 2016*; *Li et al., 2017*).

As observed in several studies (*Peláez-Peláez, Bustamante & Gómez, 2016*; *Ashraf et al., 2017*), Pb content decreased in the order soil >root >shoot ($P < 0.01$); however, 15% of the root samples had more Pb than soil.

Lead concentrations in soil and in shoots are above the maximum permitted levels, 70 mg/kg for soil and 10 mg/kg for shoots. Only 14.4% of root Pb is translocated to the shoots, so Pb enrichment in the roots would act as a barrier to heavy metal translocation, limiting its transfer to the aerial part of the grassland, as reported in *Trifolium repens* and *Lolium perenne* (*Bidar et al., 2009*), *Panicum maximum*, *Brachiaria decumbens* and *Brachiaria brizantha* (*Nascimento et al., 2014*) and in other plants (*Akinci, Akinci & Yilmaz, 2010*; *Fahr et al., 2013*; *Amari, Ghnaya & Abdelly, 2017*). Barrier mechanisms lead to the accumulation of heavy metals in the cell wall (*Sharma & Dubey, 2005*; *Samardakiewicz et al., 2012*; *Emamverdian et al., 2015*; *O'Lexy et al., 2018*).

For most plant species, the majority of absorbed lead (approximately 95%) is accumulated in the roots, and only a small fraction is translocated to aerial plant parts (*Kumar, Smita & Cumbal, 2017*). In this study, from the total Pb in the plant, 88.7% was accumulated in the roots and 11.3% was in the shoots (Table 2).

## Lead concentration in the soil

Soil Pb levels showed no difference between dry and rainy season ($P > 0.05$), coinciding with the report of *Osobamiro & Adewuyi (2015)* on agricultural soils in southwestern

Ogun State in Nigeria. Other studies find seasonal differences (*Bidar et al., 2009*; *Yabanli, Yozukmaz & Sel, 2014*), which could be due to physicochemical differences in soils (*Sharma & Dubey, 2005*).

In the study, there were also no differences in Pb content between cultivated and natural grassland soil ($P > 0.05$), as suggested by *Sharma & Dubey (2005)*.

The current result is well above the mean Pb content of the world's soils (25 mg/kg) (*Kabata-Pendias & Mukherjee, 2007*) and it seems likely that Pb in the soils of the Peruvian Andes, close to mining-metallurgical activities, is at the upper end on a global scale, as reported in several studies (*Salman et al., 2019*) WoS; (*Shi et al., 2019*; *Osobamiro & Adewuyi, 2015*; *Santos-Francés et al., 2017*; *Castro-Bedriñana, Chirinos-Peinado & Peñaloza Fernández, 2020*).

In Peru, the reference value for average Pb concentrations in agricultural soil is 70 mg/kg (*MINAM, 2017*) and in the present study (Table 2) it was observed that the average Pb concentration was 3.03 times higher than this maximum limit ($P < 0.01$), with negative implications for South American cattle, sheep and camelids and, consequently, for public and environmental health (*Chirinos-Peinado & Castro-Bedriñana, 2020*). Similarly, it was 4.25 times higher than the limit for forage cultivation recommended by *Kabata-Pendias & Pendias (2001)* of 50 mg Pb/kg of soil.

Our results are also superior to other international references that indicate a range of 7.5 to 135 mg/kg for agricultural soils in the United States (*Peláez-Peláez, Bustamante & Gómez, 2016*) and in Auckland (New Zealand) that consider a Pb content between 1.5 and 65 mg/kg (*Auckland Regional Council, 2002*; *Auckland Council, 2015*).

Soil Pb concentrations of 10–30 mg/kg do not affect plant growth (*Mlay & Mgumia, 2008*), concentrations above 30 mg/kg are considered phytotoxic to plants (*Boularbah et al., 2006b*; *Kabata-Pendias & Pendias, 2001*), and in our study the soil has 7.1 times more that the fitotoxic concentration.

Considering a reference Pb content in the Andean soils of Peru of 44.87 mg/kg, before the beginning of mining-metallurgical activities (*Santos-Francés et al., 2017*), our findings (212.36 mg/kg) are shocking, and may even have legal significance, since the emissions from mining-metallurgical companies would be altering environmental, animal and human health, and would also be causing damages to the economic systems that affect people exposed to these conditions.

The problem of contaminated soils and the products obtained from them has been neglected in Peru. In the central Andes, for many years the Pb smelter released dangerous doses of Pb emissions into the air and soil, which are transferred to the food chain (*Hou et al., 2014*), recommending that decontamination strategies and monitoring of the impact of soil contamination on crop quality be applied (*Wuana & Okieimen, 2011*) and that the mining and metallurgical industry control and reduce heavy metal emissions because they can remain in the environment for 150 to 5,000 years (*Saxena et al., 1999*).

## Lead concentration in the root

The average Pb content in the roots of the high Andean grassland was 154.65 ± 52.85 mg/kg (Table 2). Range (77.04–263.61 mg/kg) was consistent with other reports (*Zou et al., 2011*; *Pourrut et al., 2011*; *Emamverdian et al., 2015*).

This high Pb content would force plants to develop tolerance mechanisms in their root system, which would prevent further uptake and translocation to the edible parts of the plant, reducing its harmful effects (*Fahr et al., 2013*; *Amari, Ghnaya & Abdelly, 2017*; *Akinci, Akinci & Yilmaz, 2010*).

These mechanisms depend on the species of the plant (*Arshad et al., 2008*), the concentration of the metal, the duration of exposure (*Pourrut et al., 2011*), the intensity of stress, the stage of development and the type of organs and tissues of the plant.

Detoxification mechanisms include selective absorption of metals, excretion, complexation by specific ligands, compartmentalization (*Gupta et al., 2010*; *Jiang & Liu, 2010*; *Krzesłowska et al., 2010*; *Maestri et al., 2010*), the formation of calluses between the plasma membrane and the cell wall that acts as a barrier against metals (*Samardakiewicz et al., 2012*; *O'Lexy et al., 2018*), Pb sequestration in vacuole with complex formation, action of uronic acids on the cell wall, synthesis of phytochelatines and oxalates, binding with glutathione and amino acids, formation of mycorrhizae, precipitation with radical exudates, synthesis of osmolites and activation of the antioxidant defense system (*Sharma & Dubey, 2005*; *Emamverdian et al., 2015*) to eliminate reactive oxygen species (*Pourrut et al., 2011*; *Ashraf et al., 2017*).

Toxicological bioassays report biochemical and physiological actions from exposures as low as 0.1 μM Pb in spruce (*Goldbold & Kettner, 1991*), in corn $10^{-5}$M Pb (*Obroucheva et al., 1998*), or in soils above 10 mg/kg Pb (*Breckle, 1991*). *Kozlow (2005)* reports critical toxicity levels of 10 and 50 mg/kg for Pb-sensitive and moderately tolerant species; *Krämer (2010)* reports a critical level of 0.6–28 mg/kg, and *Cheng (2003)* reports changes in bean cell division with 1.0 ppm Pb.

In most plants, about 90% of total Pb accumulates in the root (*Kumar, Smita & Cumbal, 2017*). In the present study, the roots accumulated 88.7% of Pb from the plant and 11.3% was translocated to the aerial part of the grasses.

## Lead concentration in shoots

The outbreaks had 1.97 times more Pb than the limit value of 10 mg/kg for fodder (*Boularbah et al., 2006b*; *Kabata-Pendias & Mukherjee, 2007*); they exceeded more times the suggested values for vegetables, such as 3.0 mg/kg (*Robinson et al., 2008*) and 0.3 mg/kg set by the Codex Alimentarius for livestock feed (*ONU/FAO, 2018*).

The Pb content in these pastures is attributed to sustained contamination from mining-metallurgical activity in the central highlands of Peru, which permanently threatens Andean livestock, generating problems at the base of the food chain to the detriment of animal and human health and productivity (*Chirinos-Peinado & Castro-Bedriñana, 2020*).

In this study, no differences were evidenced between Pb concentrations by type of pastures and sampling periods, results that are also reported by *Orellana et al. (2019)*. The average content is higher than reported in some regions of the country and the world;

thus, in Mantaro-Jauja-Peru district, in soils with 208 mg/kg of Pb, the aerial sunflower tissues had less than 15 mg/kg (*Munive et al., 2020*). In Apata-Jauja, *Lolium × hybridum* Hausskn and *Medicago sativa* soils had $57.17 \pm 6.29$ and $50.10 \pm 8.99$ mg/kg Pb and the aerial part of these grasses had $1.17 \pm 0.69$ and $1.62 \pm 0.68$ mg/kg ($P > 0.05$) (*Orellana et al., 2019*). In Paccha-Yauli-Peru, in soils with 132–284 mg/kg Pb, the sprouts of natural and cultivated grasses had 19.5 and 20.7 mg/kg Pb (*Castro-Bedriñana, Chirinos-Peinado & Peñaloza Fernández, 2020*).

In Magdalena-Colombia, in *Brachiaria decumbens*, *B. humidícola* and *B. brizantha*, cultivated near the oil refinery, in leaves and stems are reported 2.04–3.34 and 3.16–5.36 mg/kg Pb (*Peláez-Peláez, Bustamante & Gómez, 2016*). Pb contents between 0.11 and 0.21 mg/kg are reported in cultivated pastures in New Zealand (*Longhurst, Roberts & Waller, 2004*), in those of southern New Zealand 10.6 mg/kg (*Martin et al., 2017*), in Kjeller-Norway 8.4–9.6 mg/kg (*Johnsen & Aaneby, 2019*), in the Netherlands 0.6 mg/kg *Adamse, VanderFels-Klerx & deJong (2017)* and in the central mountainous part of the Kakheti-Georgia region 0.02–0.28 mg/kg (*Bregvadze et al., 2018*).

In *Lolium multiflorum* Lam. in artificially prepared soils with 60 mg/kg Pb and different pH values and OM contents, the shoots had between 4.42–10.90 mg/kg Pb; the OM content was inversely related to the Pb content and the higher the acidic soil, the higher the Pb content (*Kwiatkowska-Malina & Maciejewska, 2013*).

In *Trifolium alexandrinum*, one of the main forage crops in Punjab-India, Pb contents between 2.83–9.17 and 2.5–4.33 mg/kg are reported in soils and shoots (*Bhatti, Sambyal & Nagpal, 2016*).

On the other hand, the Pb content of the shoots in this study was lower than that found in more contaminated soils, such as in Recuay-Ancash-Peru (3433 m a.s.l.) in soils near a polymetallic mining concession that has been operating for 70 years, the highest Pb concentration was in the *Juncus bufonius* root (718.44 mg/kg) and in the *Pennisetum clandestinum* shoot (236.86 mg/kg) (*Chang Kee et al., 2018*).

In Hualgayoc-Cajamarca-Peru, around a polymetallic mine (soils with 120 to 111,290 mg/kg Pb), the shoots of *Plantago orbignyana* Steinheil, *Lepidium bipinnatifidum* Desv., *Baccharis latifolia* Ruiz & Pav Pers. and *Sonchus oleraceus* L. had between 6,070–8,240, 6,300–7,240, 2,120–3,060 and 2,180–2,900 mg/kg (*Bech et al., 2012*).

At sites near Indian metallurgical activities, the Pb in forage was 29.06 mg/kg (*Swarup et al., 2005*), while at non-industrialized sites it was 2.08 mg/kg. In contaminated soils in Pakistan, average Pb concentrations between 36.85 and 60.21 mg/kg are reported for *Trifolium alexandrium*, *Brassica campestris* and *Avena sativa* grasslands (*Iqbal et al., 2015*).

In Ibadan-Nigeria, in soils contaminated with Pb-slag, grasses were between 209 and 899 mg/kg (*Ogundiran et al., 2012*). In Liaoning-China, in soils with 1,560 mg/kg Pb) the Pb concentration in *Lolium perenne* sprouts was 34.38 mg/kg (*Zhang et al., 2017*) and in another study in artificially contaminated soils with 500–1,500 mg/kg Pb, *Lolium perenne* sprouts had more than 50 mg/kg (*Zhang et al., 2019*).

One aspect to consider is that, although Pb from the air could enter the plant through the leaves, it has been established that most of it is absorbed from the soil and accumulates in the roots from where it moves to the shoots (*Sharma & Dubey, 2005*; *Kumar, Smita*

*& Cumbal, 2017*; *Wu et al., 2020*). In field studies evaluating the bioaccumulation and translocation of heavy metals, no reference is made to their possible absorption from the air through the leaves, so little is known about this complex and important issue, which would be influenced by the morphological and histological characteristics of the leaves and the size, load and composition of the particles emitted (*Sharma et al., 2020*).

## Lead transfer in the soil-root-shoot system

The Pb content of the soil is transferred and bioaccumulates in the edible parts of the forage, being a direct route for its incorporation into the food chain, causing damage to soil microorganisms, plants, animals and humans (*Tóth et al., 2016*; *Kryshna & Mohan, 2016*; *Hou et al., 2014*; *Chirinos-Peinado & Castro-Bedriñana, 2020*).

The bioconcentration factor (BCF) of Pb from soil to roots was $0.74 \pm 0.26$ (range: 0.31–1.43 mg/kg) (Table 3). The 17.4% of the pasture samples had a BCF >of 1, a result that indicates that Andean grasses could be considered moderate accumulators of Pb (*Chang Kee et al., 2018*; *Zhang et al., 2019*; *Wu et al., 2020*), since their roots absorb and bioconcentrate a large part of the Pb present in the soil and retain it (*Sharma & Dubey, 2005*; *Bech et al., 2012*; *Kumar, Smita & Cumbal, 2017*; *Wu et al., 2020*), transferring it to a lesser extent to the aerial part of the plant; however, the grassland shoots evaluated have a high Pb content for animal feed (*ONU/FAO, 2018*).

The determination of the bioaccumulation factor (BF) of heavy metals in the sprouts in relation to the total soil content is an appropriate method to quantify the bioavailability of the metals, and also reflects the capacity of the plants to mobilize and capture the heavy metals. Pb has a lower bioaccumulation factor (BF) than other metals because it is bound to soil colloids and is less bioavailable to the plant (*Violante et al., 2010*) and in this study its value was lower than the average recorded for spinach, lettuce, carrots, and radishes grown in soils containing 100 mg/kg Pb (*Intawongse & Dean, 2008*), which would indicate that pastures bioaccumulate less Pb than vegetables.

The translocation factor (TF) of Pb from root to shoot of the evaluated grasses ratifies the cumulative action of the root (*Kumar, Smita & Cumbal, 2017*; *Wu et al., 2020*) by translocating a low percentage of lead to the shoot.

## Implications for central Peru

The results contribute to the current knowledge about Pb transfer in the soil-root-plant system in the grasslands of the Andean region, a topic little explored in this part of the world where metallurgical mining activity emits particulate material loaded with lead and other heavy metals since 1922; therefore, it is a critical problem that must be solved.

Our results agree with Du and others (*2020*) that the main source of heavy metals is the mining and smelting operations.

The data from this study show that agri-food production on Pb-contaminated substrates in areas close to mining-metallurgical activity is a serious public health problem, and if the mining-metallurgical industry does not improve its production processes and if strict environmental adaptation programs are not implemented, the rate of chronic Pb bioaccumulation could continue with serious effects especially on vulnerable populations

(children, pregnant mothers and babies), leading to undesirable economic and social outcomes in exposed populations.

A new Peruvian guideline on the levels of Pb and other heavy metals in soil, feed, and food could help correct this problem and minimize the adverse effects of contamination.

## CONCLUSIONS

This research has revealed that the mining-metallurgical activity developed in central Peru during almost a century has had an impact on the high concentration of Pb in the upper soil layer and in the natural and cultivated pastures of the central highlands of Peru.

Lead in soil was 3.04 times higher than the maximum limit for agricultural soil and in forage was 1.97 times higher than the limit value for forage, and of the total Pb content in pastures about 90% accumulates in the roots.

The study provides comparable information on the content and transfer of Pb in the soil-root-plant system of pastures used to feed livestock raised in areas close to polymetallic mining-metallurgical activity in the high central Andes of Peru. The study is one of the first in this part of the world, and the conclusions are shocking and could lead to socio-economic problems in exposed populations.

## ACKNOWLEDGEMENTS

The authors are grateful to engineer Surveyor Cochachi, director of the Yauli-La Oroya Agricultural Agency, for his contact with Peasant Paccha Community. We also thank the editor and reviewers for their valuable suggestions and comments that have improved this scientific article.

### Funding

This work was supported by funding award from mining fees and royalties of the General Research Institute of the Universidad Nacional del Centro del Perú (No 004-2017-VRI-UNCP). The funders had no role in study design, data collection and analysis, decision to publish, or preparation of the manuscript.

### Grant Disclosures

The following grant information was disclosed by the authors:
General Research Institute of the Universidad Nacional del Centro del Perú: 004-2017-VRI-UNCP.

### Competing Interests

The authors declare there are no competing interests.

### Author Contributions

- Jorge Castro-Bedriñana conceived and designed the experiments, analyzed the data, prepared figures and/or tables, authored or reviewed drafts of the paper, and approved the final draft.

- Doris Chirinos-Peinado conceived and designed the experiments, performed the experiments, analyzed the data, prepared figures and/or tables, authored or reviewed drafts of the paper, and approved the final draft.
- Edgar Garcia-Olarte and Rolando Quispe-Ramos performed the experiments, analyzed the data, authored or reviewed drafts of the paper, and approved the final draft.

## Field Study Permissions

The following information was supplied relating to field study approvals (i.e., approving body and any reference numbers):

Sampling of soil, roots and shoots from cultivated and natural pastures was approved by the President of the Peasant Community ''Purísima Concepción de Paccha'', Yauli, La Oroya, study which is part of a larger project (project number: 1565-R-2917-UNCP).

## Data Availability

Raw data are available as a Supplementary Files.

## Supplemental Information

Supplemental information for this article can be found online at http://dx.doi.org/10.7717/peerj.10624#supplemental-information.

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
