# Peer review of "Lead transfer in the soil-root-plant system in a highly contaminated Andean area"

_PeerJ, doi:10.7717/peerj.10624_

## Round 0.1 · original submission · Major Revisions

Both reviewers were generally complimentary of your research. They have made some good comments that when addressed will improve the clarity of your manuscript. I would also emphasize that your revised manuscript should be edited for grammar. It is important that you pay particular attention to this. I cannot accept a revised manuscript that hasn't been checked for proper grammar.

·

Basic reporting

I consider this manuscript of interest because it contributes to current knowledge on Pb transfer in soil-root-shoot systems in the Andean region, a seldom explored topic in this part of the world.
I strongly suggest the authors to find editorial assistance. The style and writing needs a major review and more care and prolixity to avoid grammatical and concordance mistakes, which already makes the article difficult to read in some sections. In addition, the article has very long sentences (e.g. L6-L11), which makes it difficult to follow the ideas.
Although the article has a good grasp of relevant and recent literature, the authors fail to support statements with appropriate references, and in some cases they do not use them correctly.
The structure of the article is disorganized, there is no clear hypothesis, information is repeated in different parts, and tables and figures contain the same information.

Experimental design

This article meets the aims and scopes of the journal.
The research question is not well defined, although the topic is important and deserves attention given the lack of information on metal soil polluiton and translocation to grass tissues in the Andean region.
Authors should explain why they compare between natural and cultivated pastures, and two seasons in the year. There is no hypothesis to support this experimental design.
Methods do not describe pH and organic matter analysis, and do not mention brand/model of equipment used.

Validity of the findings

All data have been provided.
A strong concern is that results are not compared to reference or background Pb soil levels in Peru, nor to widely used international standards.
Although conclusions are limited to supporting results, they do not report on the dynamics of Pb transfer in the soil-root-shoot system as proposed in the research question.

Additional comments

I have the following observations and suggestions:

INTRODUCTION
L31. Toxic for what biological system?
L33. Compared to what standard or reference scale: wildlife, ecosystems, human health? Methylmercury, mercury, chromium VI, cadmium, arsenic, organochloride compounds, among others, are much more toxic and dangerous.
L51-53. I recommend the authors Chang Kee et al 2018 (Accumulation of heavy metals in native Andean plants: potential tools for soil phytoremediation in Ancash (Peru). Environ Sci Poll Res 25(34), 33957-33966. (10.1007/s11356-018-3325-z))
L60. Says 1992. Should say 1922.

METHODS
L93. Avoid Latitude and Longitude
L 93. Include m “above sea level”, and then only refer as “m a.s.l.”
L96. Repeats information (1922) given in the introduction. Avoid.
L100. Repeats information given in the introduction about the degree of pollution. Avoid.
L108-109. Should be included under Laboratory Analysis.
L114-L118. Should be included in Figure 1 legend.
L125. Reduce number or references.
L133. How was pH and organic matter analysed? Give methods and brand/model of equipment used. Also for FLAA.
L149. Authors must refer also to the National Soil Environmental Quality Standard. Why using a Standard from Finland as an international reference? There are others commonly used, such as US EPA, the Canadian Environmental Protection Act Registry, and the Dutch RIVM Standards.
L155-L164. The authors use the transfer factor incorrectly. Hu et al., 2017 mention, “this factor represents the potential capability of heavy metals’ transmission from soil to the edible parts of vegetable”. In the present work, this would only be represented by TBss = [Pb shoot]/[Pb soil]. However, the authors also use it to describe the translocation root/soil and shoot/root. In this case, I recommend the application of the bioconcentration factor (BCF) (Yoon et al. 2006), and translocation factor (TF) (Zu et al. 2005), which can be calculated using the following formulas:
BCF = metal contents in root tissue/metal contents in soil
TF = metal concentration in shoot tissue/metal concentration in root tissue
References
Yoon J, Cao X, Zhou Q, Ma LQ (2006) Accumulation of Pb, Cu, and Zn in native plants growing on a contaminated Florida site. Sci Total Environ 368:456–464. https://doi.org/10.1016/j.scitotenv.2006.01.
Zu YQ, Li Y, Chen JJ, Chen HY, Qin L, Christian S (2005) Hyperaccumulation of Pb, Zn and Cd in herbaceous plants grown on lead–zinc mining area in Yunnan, China. Environ Int 31:755–762. https://doi.org/10.1016/j.envint.2005.02.004
L166. If analyses include the effect of type of pasture and sampling period, two-way ANOVA should be performed.

L173. RESULTS
Authors should develop in more extent this section and not merely refer to tables and figures.
The information in Tables 2 and 3 is the same as in Figures 2 and 3, respectively. This should be avoided. I suggest to use Tables.
There are no results for soil composition.
L 196. This is part of the Discussion section.

DISCUSSION
L230-231. This quote is something the authors do not know or have not proven for the case of La Oroya. I suggest they mention it as a possibility but not as a fact, unless they provide evidence of this biomagnification process.
L232. This relationship between Pb in the air and the accumulation chain proposed is not clear.
L234. This is obvious and is not a finding of this work. Polluted soils will always have more metal concentration than plant tissues.
L237-238. An explanation and reference are needed to support this statement.
L241-243. A reference is needed for 25 mg/kg as the world average concentration. Authors should compare Pb soil concentrations found in this study with polluted sites around the world to state that soils of the central Peruvian highlands are at the upper end on a global scale.
L255-258. It is crucial that the authors mention soil Pb background levels for Peru in order to assure the extent of impact of the metallurgic activities in La Oroya on soil quality and its potential negative effect on environmental and human health.
L266. Are considered toxic for what biological system: plants, animals, humans, the environment?
L268. Morera et al., 2013 report on suspended sediment yield and hydrology of Andean rivers, not in soil pollution. Authors must be careful when using others’ results to support their findings.
L286. Indicate Pb concentrations under which these mechanisms to attenuate the absorption and transfer to the aerial parts were observed. In addition, mention phytostabilizing strategies in plant roots based on exudates and polymers.
L320. Indicate Pb levels in other regions mentioned.
L364-368. Should be moved to “Implications for central Peru”.
FIGURE 1.
The legend is not clear about the significance of shapes (square, circle) and colours regarding period (April/September) and origin (cultivated/natural).
Avoid picture of the city. It does not provide useful information. Replace image for sampling sites with a black/white map showing distance from La Oroya city.

Reviewer 2 ·

Basic reporting

no comment

Experimental design

- Line 134 : What the authors mean by normal temperature? the usual procedure for plant material drying is 60-80°C for 48h or till a constant weight was obtained. This is supported by many published works for example: Maryland, 1968, Agron. J. 60, 658-9; Sharkey 1970 Grass Forage Sci. 25, 289-294; Ramsumair et al., 2014.Trop Agric. 91,179-86; Cone et al., 1996. J Ag Sci. 126, 7-14).

- For laboratory analysis, the authors wash the roots and shoots with deionized water. This washing is poorly appropriate and does not eliminate the metals adsorbed on the surfaces as well as the Pb deposits on the shoots that can increase the Pb level recorded in the aboveground part of the plant. In addition to washing plant material with water, many authors rince roots and shoots using cold 0.2 mM CaSO 4 solution (auguy et al 2013). this makes it possible to eliminate the Pb adsorbed on the surface of the organs and makes it possible to really estimate the metal absorbed by the plant

Validity of the findings

- Line 372 : “Our findings are important and impressive and demonstrate that the grasslands in the central highlands of Peru would have high concentrations of heavy metals compared to other parts of the country and the world…….”. There is no data in this work that support such confirmation. This study is focused on lead and it is difficult to generalize to other heavy metals.

- In this work, it would be knowledgeable if the authors provided informations supporting that the Pb analyzed in the aerial part of the plants comes from the soil and that the Pb from the air do not overestimate the results.

Additional comments

Lead is a highly toxic heavy metal that affect plants growth and yield and impact grass based forage quality in many natural and cultivated pastures areas. This paper provides new data and interesting point of view on this topic.
There still some questions and points to clarify

- Figure 1 : The study site and sampling location are 20 kilometers from the lead extraction site. To better estimate the impact of La Oroya Mining-Metallurgical complex on the site contamination, I propose to indicate the orientation of the sampling site and the direction of the wind.
On the map, I suggest to eliminate the arrows and to put a square indicating the studied area.
The images also show a river that may contribute to the transport of metal contaminants. If it is the same river, indicate the direction of the water flow.

- Line 59 and 60: it seems there is an error in the date. Put “The polymetallic mining-metallurgical center located in the central sierra, which has been in operation since 1922” (as indicated in line 96) instade of “The polymetallic mining-metallurgical center located in the central sierra, which has been in operation since 1992 ”

- It appears that the study area is subject to emissions of particulate matter that contaminate the surrounding ecosystems. This source of Pb is described in the introduction and the authors are well aware of it (lines 61, 62 and 63). However, they have not integrated this aspect into their experimental approach and did not take this into account in the analysis and discussion of the results.

- In discussion of lead concentration in the soil, the authors give no reference to mobile and mobilizable fractions of Pb. This is important because the Pb is poorly soluble and a very small part of the metal is really bioavailable to plants. In practice, such fractions can be determined by quantification of Pb in CaCl2 and EDTA extractable fractions (Auguy et al 2013).

- Line 290 and 291 : this sentence is not clear

- Line 319 : the authors claim that “In the current study, no differences were found between Pb concentrations in natural and cultivated pastures”. I think this is mostly due to Pb deposit from air emission Pb particulate matter.

- Line 337 : please put 26.8 mg/kg instead of 26.8 mg/g

- Line 340 : “In practice, accumulation of Pb in forage roots instead of shoot are desirable… “ will be more appropriate for this sentence

- Lines 355 to 357: BF also reflect the capacity of plants to mobilize and uptake heavy metals.

---

## Round 0.2 · Minor Revisions

Improvements have been made to your revised manuscript, however, there are still issues with grammar and clarity. Your results are interesting and novel, but the presentation of the work still needs improvement.

·

Basic reporting

Although there has been some improvement, I’m afraid the manuscript still does not meet the standards of a scientific article in terms of writing and English style. I think the information is worth considering, especially because it covers a seldom explored topic in the Andean region. However, it needs to be presented in a more organized way, in particular in the Introduction and Discussion. These sections need to be restructured. Ideas do not follow a consistent path, from general to specific topics, which makes it difficult to read. Also, the manuscript still has very long sentences (e.g. L62-L68, L79-L85, L86-L92), which is not helpful to follow.

Experimental design

Still is not clear for me what is the hypothesis behind sampling native and cultivated grasses. Also, it would have been very interesting if the authors analysed metals in each grass species. This may provide important information on bioaccumulation profiles in edible parts, and hence, the potential risk of each grass species to animal (and human) health.
L102. I meant to avoid the terms “Latitude” and “Longitude”, not the information on coordinates.
L102. Include m “above sea level” for the first time, and then only refer as “m a.s.l.”
L116-118. Should be included under Laboratory Analysis. This was noted and suggested in the first review.
L149. Include reference to the Walkley-Black method and equipment used.
L189. If analyses include the effect of plant species and sampling period, a two-way ANOVA should be performed to check for interactions between these two factors.

Validity of the findings

There are no results for soil composition although calculation of percentages of sand, silt and clay are mentioned under Laboratory analysis. I noted this in my first review.
- Discussion
L258. This quote should explicit that this is a possibility. I suggest “which would be deposited in the soil and most likely assimilated by plants and animals…”
L260. Include Kryshna and Mohan, 2016 in the list of references.
L266. A brief explanation to this barrier mechanism should be included.
- Figure 1.
The images are not clear enough for a publication. Replace image of sampling sites with a black/white map showing distance from La Oroya city.

Additional comments

Comments are already included in the above sections.

Reviewer 2 ·

Basic reporting

no comment

Experimental design

no comment

Validity of the findings

no comment

Additional comments

The revised version of the article is consistent with the observations and comments. Several sections of the manuscript have been really improved.
The maps in Figure 1 can be improved. The position of the studied site is still not clear and the the writing in the maps is very small.

---

## Round 0.3 · accepted · Accept

Thank you for your efforts to improve your manuscript's readability.